# Synthesis and Peroxide Activation Mechanism of Bimetallic MOF for Water Contaminant Degradation: A Review

**DOI:** 10.3390/molecules28083622

**Published:** 2023-04-21

**Authors:** Mengke Fan, Jingwei Yan, Quantao Cui, Run Shang, Qiting Zuo, Lin Gong, Wei Zhang

**Affiliations:** 1School of Ecology and Environment, Zhengzhou University, Zhengzhou 450001, China; 2School of Water Conservancy and Civil Engineering, Zhengzhou University, Zhengzhou 450001, China; 3Henan Key Laboratory of Water Pollution Control and Rehabilitation Technology, Faculty of Environmental and Municipal Engineering, Henan University of Urban Construction, Pingdingshan 467036, China; 4Henan International Joint Laboratory of Water Cycle Simulation and Environmental Protection, Zhengzhou 450001, China; 5Zhengzhou Key Laboratory of Water Resource and Environment, Zhengzhou 450001, China; 6Yellow River Institute for Ecological Protection and Regional Coordination Development, Zhengzhou University, Zhengzhou 450001, China; 7Henan Key Laboratory of Water Resources Conservation and Intensive Utilization in the Yellow River Basin, Zhengzhou 450046, China

**Keywords:** bimetallic MOFs, advanced oxidation process, catalyst, degradation

## Abstract

Metal–organic framework (MOF) materials possess a large specific surface area, high porosity, and atomically dispersed metal active sites, which confer excellent catalytic performance as peroxide (peroxodisulfate (PDS), peroxomonosulfate (PMS), and hydrogen peroxide (H_2_O_2_)) activation catalysts. However, the limited electron transfer characteristics and chemical stability of traditional monometallic MOFs restrict their catalytic performance and large-scale application in advanced oxidation reactions. Furthermore, the single-metal active site and uniform charge density distribution of monometallic MOFs result in a fixed activation reaction path of peroxide in the Fenton-like reaction process. To address these limitations, bimetallic MOFs have been developed to improve catalytic activity, stability, and reaction controllability in peroxide activation reactions. Compared with monometallic MOFs, bimetallic MOFs enhance the active site of the material, promote internal electron transfer, and even alter the activation path through the synergistic effect of bimetals. In this review, we systematically summarize the preparation methods of bimetallic MOFs and the mechanism of activating different peroxide systems. Moreover, we discuss the reaction factors that affect the process of peroxide activation. This report aims to expand the understanding of bimetallic MOF synthesis and their catalytic mechanisms in advanced oxidation processes.

## 1. Introduction

Emerging organic contaminants (EOCs), including persistent organic pollutants, pharmaceuticals, personal care products (PPCPs), and endocrine disruptors (EDCs), are a group of compounds that are characterized by their persistence, bioaccumulation potential, and endocrine-disrupting properties [1]. EOCs typically enter the environment through human excretion or disposal processes, posing a continuous ecological risk to aquatic organisms and humans alike. Therefore, the removal of EOCs from water has become a major concern among scientists [2,3,4].

The complex composition of EOCs in the aquatic environment coupled with their resistance to biodegradation renders conventional physicochemical methods ineffective. In light of this, advanced oxidation processes (AOPs) have gained widespread attention as an alternative strategy. AOPs are known for their high oxidation efficiency and lack of secondary pollutant generation, making them an attractive option for the removal of EOCs from water [5,6].

AOPs are a cost-effective technology used to rapidly degrade non-biodegradable substances in water [7,8]. AOPs involve various oxidation mechanisms, including photocatalytic oxidation, electrocatalytic oxidation, and Fenton-like oxidation, which have been widely utilized and shown to effectively remove non-biodegradable substances from water [9,10]. The AOPs involves the generation of reactive oxygen species such as •OH, SO_4_^•−^, ^1^O_2_, and others, which can be utilized to treat wastewater. Free radicals possess a high redox potential and non-selective oxidation, allowing them to easily oxidize other molecules while reducing themselves. These reactive oxygen species are crucial in initiating AOPs because they convert complex toxic pollutants into simple, non-toxic substances [11,12]. Among them, SO_4_^•−^ is the most reactive and has been observed to attack a wide range of organic contaminants. In aqueous solutions, SO_4_^•−^ has a redox potential of 2.5–3.1 V, a broad pH range adaptation, and a long half-life [13,14]. The activation of peroxide is of great significance, and numerous scholars have conducted in-depth research on catalysts.

MOFs are highly regarded peroxide catalysts that consist of metal ions self-assembled with organic ligands through coordination bonds [15,16]. Their tunable energy band structure, large specific surface area, high porosity, and intrinsic catalytic activity are of the reasons for their significant research interest in adsorption and catalysis [17]. Furthermore, their modification ability after synthesis and structural properties support the vast research efforts in this field, which may lead to various commercial applications. However, MOFs have limitations in the catalytic process such as low electron conductivity, a single type of unsaturated metal site, and easy inactivation during the reaction [18]. Compared to other inorganic catalysts, MOFs can be designed and microfabricated at the atomic level, allowing their structures and functions to be adjusted via methods such as doped metals, modified ligands, and loaded carbon-based materials [19,20]. Among them, the bimetallic synergy produced by MOF-doped metals is considered to be one of the most effective strategies to achieve enhanced catalytic activity and/or expand the reaction range [21]. Novel composite materials with bimetallic MOFs have gained considerable attention in heterogeneous catalysis. This approach preserves the MOF’s skeleton while enhancing cycling stability and interfacial electron transfer efficiency [22].

Although single-metal MOFs offer a wide range of structural compatibilities, the incorporation of transition metals (Fe, Co, and Ni) into MOFs as bimetallic composites can reduce cost and further improve catalytic activity [23,24]. Bimetallic MOFs can be categorized into two types of spatial arrangements: different metals forming different secondary-building units (SBUs) or different metals fixed in the same SBUs. The latter exhibits higher catalytic activity due to its compact structure, better stability, and electron transfer efficiency [25]. The combination of two different metal cations can enhance conductivity and enable oxidation reactions between different metal sites in the MOF’s structure, leading to increased catalytic efficiency. This controllable integration of functional components can construct a multifunctional complex with advanced properties, enhancing activity for redox catalytic reactions, supercapacitors, and other reactions [26,27]. While the development of bimetallic MOFs is still in its infancy, an increasing number of studies demonstrate their great potential in various practical applications [28].

This review systematically introduces the preparation methods of bimetallic MOF composites, the synergy between bimetallic sites of bimetallic MOFs, and their application prospects in emerging fields to demonstrate their structural advantages. Finally, the paper provides an outlook on the challenges and future prospects of the synthesis and application of bimetallic MOFs to provide insights for the rational design of hybrid MOFs with complex structures and fine functions.

## 2. Preparation of Bimetallic MOFs

Due to the inherent advantages of bimetallic MOFs, considerable efforts have been made to synthesize such materials. Generally, methods for constructing bimetallic MOFs can be broadly classified into two categories (Figure 1). The first involves direct synthesis of bimetallic MOFs using techniques such as hydrothermal/solvothermal [29] and microwave-assisted methods [30] or preparation of core–shell bimetallic MOFs by controlling the nucleation and growth kinetics of guest and host MOFs. The second category comprises a stepwise synthesis method in which monometallic host MOFs are first synthesized and then doped with metals using ion exchange or seed induction to form bimetallic MOFs. The composition and structure of bimetallic MOFs are determined by parameters such as the crystal structure of the MOFs, the size of the doped metal ions, the lattice matching of the MOFs, and the chemical stability of the MOFs. The mechanisms involved in these representations are discussed in detail below through typical examples. Detailed strategies for synthesizing bimetallic MOF hybrids are discussed in the following sections.

### 2.1. One-Step Synthesis

The one-step synthesis method is a commonly employed approach for synthesizing bimetallic MOFs. However, due to the differing binding kinetics of the two metal ions to the ligand, it can be challenging to predict the resulting topology of the metal–ligand framework via this method. To construct solid-solution bimetallic MOFs with desired properties, precise control of the reaction conditions is crucial. For instance, the molar ratio of the two metal sources, reaction time, solubility of metal ions, and pH of the reactant solution all have significant impacts on the structure and morphology of the resulting bimetallic MOFs [31]. The synthesis of bimetallic MOFs typically involves the reaction between ligands and two metal ions with nearly identical electronic configurations and charge densities. Following extensive exploration by numerous researchers, this method has become well established, resulting in the successful synthesis of a diverse array of bimetallic MOFs.

#### 2.1.1. Hydrothermal/Solvothermal Method

Hydrothermal/solvothermal synthesis is a commonly used method for preparing MOFs. This approach involves the dissolution of organic ligands and metal complexes in water or other organic solvents followed by the reaction of the mixture under high-temperature and high-pressure conditions [32,33]. Significant progress has been made in the synthesis of bimetallic MOFs through the efforts of numerous scientific research groups. For example, Co^2+^ and Zn^2+^ ions were dissolved in methanol solution with 2-methylimidazole to synthesize Co_x_·Zn_1−x_ (MeIm)_2_ [34]. This method offers high crystallinity and controllable crystal size, enabling the synthesis of well-oriented and perfect crystals. Additionally, a variety of organic functional groups and metal complexes can be introduced to prepare customized MOFs [35]. Wang et al. incorporated a series of transition metals (M = Co, Ni, and Zn) into Fe-BPTC to form three different bimetallic MOFs: Fe_2_Co-BPTC, Fe_2_Zn-BPTC, and Fe_2_Ni-BPTC [36]. Other examples include the solvothermal synthesis of Ce/Zr UiO-66 by mixing cerium (III) chloride and zirconium chloride [37] and the hydrothermal synthesis of Ni/Co-MOF by loading Ni^2+^, Co^2+^, and 1,4-phthalic acid organic ligands into a polytetrafluoroethylene-lined stainless steel autoclave [38]. Wang et al. doped Ni-MOF with Fe^3+^ to form FeNi-MOFs, which were analyzed spectroscopically to reveal the successful replacement of part of the Ni with Fe^3+^ to form two types of SBUs that were uniformly distributed in MOF crystals [39]. Similarly, Li et al. synthesized Ti-In-MOF in one step using the hydrothermal method (Figure 2a), while Han et al. prepared La-Zr bimetallic MOFs adsorbents by placing La and Zr salt precursors and 2-aminoterephthalic acid ligands into an autoclave (Figure 2b) [40] These bimetallic MOFs exhibit excellent adsorption properties for As(V) or Cr(VI) in strongly acidic solutions with a maximum adsorption capacity for Cr(VI) of up to 222.5 mg/g. This method integrates the advantages of two metals to achieve intensified synergism of adsorption and photocatalytic degradation with shorter reaction times and the promotion of high-dimensional MOF structures compared to synthesis at room temperature. However, this approach generates a large amount of solvent waste and is potentially hazardous when handling metal salts in the presence of organic liquids [41].

#### 2.1.2. Microwave-Assisted Method

In the past, conventional heating was the primary energy source for MOF synthesis. However, microwave-assisted and sonochemical heating methods are now being increasingly used to provide the necessary energy for the reaction. Microwave-assisted synthesis employs electric or magnetic fields to induce high-speed collisions between charged particles, leading to the production of high-purity crystals [44,45]. As illustrated in Figure 2c, Chen et al. used the microwave method to synthesize a bimetallic solution with Co and Ni ions in the same SBUs. Sonochemistry enables rapid reaction times, production of crystals with uniform particle morphology, and isolation of products with minimal or no secondary products [42]. Some researchers have utilized fast, low-cost, and direct ultrasonic-assisted synthesis to prepare porous bimetallic Co/ZIF-8, which effectively promotes nucleation and limits particle size to the nanoscale region (Figure 2d) [43]. However, these methods have certain limitations, including the formation of crystal particles that are too small for single-shot X-ray analysis [44]. Additionally, the cost and yield of these methods are not always proportional, and the ability to control reaction conditions by varying the irradiation power, reaction time, and temperature is limited by the possibility of different instruments providing non-identical conditions, ultimately hindering reproducibility.

#### 2.1.3. Nucleation Dynamics Control Method

The current technique involves mixing the ligands and metals of two MOFs with different nucleation rates and constructing a core–shell structure by controlling the nucleation and growth rates of the two MOFs, which can synthesize core–shell bimetallic MOFs in one simple step [46,47]. The key to success is the difference in the growth rates of the two metal–ligands, and the structure and composition of the final product is also linked to the growth rate of the two components. Under the same reaction conditions, seeds with fast nucleation velocity grow as guest MOFs and are enveloped with guest MOFs with a slow nucleation speed, leading to core–shell bimetallic MOFs [48,49,50]. Nevertheless, it is challenging to create core–shell structures using a one-step synthesis method, and the synthesis process necessitates precise control of the procedure.

Yang et al. employed a one-pot synthesis method to add two linkers with high-connectivity (TCPP, H_4_TCPP = tetrakis (4-carboxyphenyl) porphyrin) and low-connectivity connectors (BPDC, BPDC = biphenyl-4,4′-dicarboxylate) to the solvent to create two crystal nucleation rates with different environments (Figure 3a). Under kinetic control, a hybrid core–shell MOF (PCN-222@Zr-BPDC) with mismatched lattices was successfully synthesized [47]. The mole percentages of the two metals impacted the nucleation kinetics of the MOF’s growth. ZIF-8 and ZIF-67 possess different reaction rates, leading to an uneven distribution of elements in the nanocrystals. Co_20_Zn_80_-ZIF grows into a core–shell structure (Figure 3b). As the proportion of Co increases, the mixed metal system gradually shifts from typical two-step growth kinetics to a trend of Co-dominated one-step growth kinetics, resulting in the growth of Co_50_Zn_50_-ZIF and Co_80_Zn_20_-ZIF into solid-solution nanostructures [50]. Another example is the utilization of one-step hydrothermal synthesis by adding Co^2+^, Ni^2+^, H_3_BTC, and PVP to the high-pressure reactor at the same time and accurately controlling deionized water:DMF (N,N-dimethylformamide):methanol = 1:1:1, leading to the successful synthesis of Ni/Co-MOF with an egg yolk shell structure (Figure 3c) [51].

### 2.2. Stepwise Synthesis Method

Designing multiple operation steps in a sequential manner is a complex yet effective method for synthesizing bimetallic MOFs with a desired structure. By using this method, metal doping can be achieved through post-synthesis ion exchange without sacrificing the crystallinity of the MOFs. In addition, seed-mediated methods can be employed to prepare core–shell bimetallic structures, which can significantly alter the physical and chemical properties of the MOF framework.

#### 2.2.1. Ion Exchange Method

MOFs can be synthesized using metal ion exchange, which involves doping metals into the MOF framework by exploiting the different affinities of metal ions and ligands. However, during the exchange of ligands or metal ions, it is essential to carefully select suitable ligand–metal combinations to prevent framework collapse. This method exhibits a certain degree of randomness and cannot accurately predict the coordination environment of the metal center. Additionally, the efficiency of doping metal ions to replace the original metal center is sometimes low because metal ions with different valence states and ionic radii tend to adopt different coordination numbers and environments. Improper metal node replacement may damage the intrinsic structure of the MOF and even cause it to rupture [28]. The speed and extent of the exchange process are influenced by the coordination number of the metal–ligand, the radius of the metal ion, and the solvent [52]. Generally, MOFs with high coordination numbers in SBUs can be synthesized using the cation exchange method [53].

For example, Cheng et al. synthesized CuCo-MOFs using the ion exchange method. As shown in Figure 4, they first obtained tannic acid (TA) chelated cobalt complex nanoboxes (TA-Co NBs) from initial ZIF-67 nanoparticles by using a chemical etching process. Subsequently, the obtained TA-Co NBs were transformed into Cu-modified TA-Co NBs (TA-CoCu NBs) in a cation exchange process using a Cu^2+^ ion solution in which some of the Co sites of TA-Co NBs were replaced by Cu atoms. Finally, TA-CoCu NBs were transformed into CoCu-MOF NBs through a ligand exchange process. During this process, the TA linkers of TA-CoCu NBs were gradually replaced by 2,3,6,7,10,11-hexahydroxytriphenyl (HHTP) organic ligands, possibly due to the stronger chelating ability of HHTP ligands compared to TA molecules [54].

#### 2.2.2. Seed-Induced Growth Method

The stepwise synthesis process involves depositing crystal materials on a specific crystal plane of a substrate with the same orientation and similar lattice spacing [55]. The substrate is first synthesized as the host MOFs followed by the epitaxial growth of guest MOFs on the interface with matching plane direction and lattice distance to obtain the bimetallic MOFs. The successful epitaxial growth of core–shell structures depends on the selection of appropriate host and guest MOFs [56], enabling relatively controllable preparation of mixed-metal MOFs [57]. The stepwise synthesis method enriches the composition of MOFs (such as ligands and/or metal centers) as well as their structural diversity (such as pores, surface properties, and functions) [58,59,60].

For instance, Qi et al. synthesized Ni-MOF via the hydrothermal method and then doped Sn to Ni-MOFs using the ion exchange method to form Sn/Ni-MOF core–shell structures (Figure 5a). This method enabled precise control of the doping metal content [61]. Zn/Co bimetallic MOFs were prepared via epitaxial growth of ZIF-8 and ZIF-67 with similar unit cell parameters (Figure 5b) [62]. Some researchers used the seed-mediated method to grow the ZIF-8@ZIF-67 core-shell structure on GO. In this method, ZIF-8 seeds were first grown in situ on GO flakes followed by the deposition of ZIF-67 crystals on the surface of ZIF-8 seeds, resulting in the formation of the core–shell structure of ZIF-8@ZIF-67 (Figure 5c) [63]. Tang et al. also employed the seed-mediated growth method to deposit ZIF-67 on the ZIF-8 crystal plane, ultimately forming the core–shell MOF (ZIF-8@ZIF-67) crystal with ZIF-8 as the core and ZIF-67 as the shell (Figure 5d) [64].

The various synthesis conditions employed in the preparation of bimetallic MOFs have a significant impact on the crystal nucleation and growth states, thereby influencing the morphology and topology of the resulting products. Furthermore, the preparation method has a substantial influence on the species of reactive oxygen species, surface oxygen vacancies, surface chemical composition, and state [65]. These physical and chemical properties in turn affect the catalytic performance of the catalyst. Additionally, it is important to note that different synthesis methods can lead to significant differences in the specific surface area and pore volume of the same material [66].

## 3. Fenton-like Reaction of Bimetallic MOFs

Compared to monometallic MOFs, bimetallic MOFs exhibit unique synergistic effects between two distinct metal elements, leading to higher stability and catalytic efficiency. The Fenton-like reaction of bimetallic MOFs and persulfate-based advanced oxidation processes have found widespread application in water treatment to degrade emerging pollutants [67]. In their research, He et al. [10] suggested that the energy efficiency of a synergistic catalytic system is higher than that of a single system. The addition of catalysts to a synergistic catalytic system can leverage the activation properties of the catalysts to effectively convert peroxides into free radicals, which can lead to efficient pollutant degradation and improved energy efficiency. Bimetallic MOFs have been identified as highly effective catalysts for peroxides due to their unique structural and chemical properties. The most commonly used peroxides in this degradation system are PDS, PMS, and H_2_O_2_, whose molecular structures are shown in Figure 6a. Table 1 provides a summary of the reaction mechanisms involved in the activation of three distinct peroxides by bimetallic MOFs. Bimetallic MOFs activate these three oxidants to generate various reactive oxygen species (ROS), which attack and degrade pollutants into smaller molecules (Figure 6b). The following section provides a detailed explanation of how bimetallic MOFs synergistically enhance the activation of different oxidants and degrade various emerging pollutants.

### 3.1. Bimetallic MOF for H_2_O_2_ Activation

H_2_O_2_ is a commonly used oxidant in wastewater treatment [68]. H_2_O_2_ can synergistically produce more •OH with the catalyst, thus promoting the degradation of antibiotics [9]. The activation of H_2_O_2_ by MOFs as a heterogeneous catalyst is referred to as Fenton-like oxidation in the field of wastewater treatment [69,70].

Pan et al. [9] investigated the activation of H_2_O_2_, which led to the production of non-selective hydroxyl radicals as well as ^1^O_2_ resulting from the oxidation of O_3_ in the system. Their experiments confirmed the presence of synergistic interactions among •OH, ^1^O_2_, and •O_2_^−^ within the system. However, excessive amounts of H_2_O_2_ can lead to self-decomposition and consumption of •OH, which can ultimately hinder the degradation of antibiotics. He et al. prepared a bimetallic MOF by introducing Co(II) into MIL-101 (Fe) via a hydrothermal method. The CIP removal rate of MIL-101 (Fe, Co) (97.8%) was found to be higher than the sum of MIL-101 (Fe) (55.2%) and MOF (Co) (14.3%). To investigate the synergistic effect between Fe and Co sites in MIL-101 (Fe, Co), the authors analyzed the material from multiple perspectives. As shown in Figure 7a,b, doping cobalt ions in MIL-101 (Fe) can reduce the corresponding activation energy barriers of H_2_O_2_ on the surface of the material from 0.55 eV to 0.40 eV, indicating that MIL-101 (Fe, Co) has a stronger activation effect on H_2_O_2_ molecules. Figure 7c illustrates the activation of H_2_O_2_ by MIL (Fe, Co). In MIL (Fe, Co) materials, Fe/Co is connected to the benzene ring in the organic ligand via the C-O-Fe/Co bond, which not only facilitates electron transfer in the coordination system but also increases the reactive active sites. During the degradation process, the benzene ring connected by MIL (Fe, Co) forms a Π–Π conjugation with the benzene ring in ciprofloxacin (CIP), promoting the adsorption of CIP by MIL (Fe, Co) and electron transfer to oxidize and decompose CIP. The electrons will be transferred from the electron-deficient center around the benzene ring to the electron-rich center around the metal for the reduction of Co(III). This step solves the key rate-limiting problem in the Fenton system and explains why the catalytic degradation efficiency of MIL (Fe, Co) for CIP is higher than that of MIL (Fe) [71].

In the CUMSs/MIL-101 (Fe, Cu) system, the coordinatively unsaturated metal sites (CUMSs) containing Fe(II)/Fe(III) with a mixed valence have a mesoporous structure with abundant open active sites. Additionally, Cu(II)/Cu(I) CUMSs can accelerate the valence cycle of the metal center, reduce the activation energy barriers of H_2_O_2_ on the catalyst surface (0.42 eV for CUMSs/MIL-101 (Fe) and 0.27 eV for CUMSs/MIL-101 (Fe, Cu) (Figure 7d,e), and further improve the catalytic activity and H_2_O_2_ utilization efficiency [72]. The mechanism of the CUMSs/MIL-101 (Fe, Cu)/H_2_O_2_ degradation of CIP is shown in Figure 7f.

Figure 8a shows the construction of CuFe_2_O_4_@MIL-100 (Fe, Cu) with a core–shell structure by Shi et al. using an in situ derivatization strategy. When stimulated by visible light, MCuFe-MOF generates photogenerated electrons and holes, which can decompose H_2_O_2_ into ·OH to degrade pollutants. Figure 8b shows the catalytic performance test results, which indicate that the catalyst not only reduces the amount of H_2_O_2_ needed in the reaction process but also enhances structural stability by reducing Fe^III^ through electron transfer. The MCuFe-MOF/ H_2_O_2_ system continuously generates ·OH through the redox interaction between ≡Fe^II^/≡Fe^III^ and ≡Cu^II^/≡Cu^I^, thereby ensuring the progress of the degradation reaction (Figure 8c). The core nanoparticles’ photothermal effect generates “hot electrons,” which can accelerate the separation of photogenerated electron–hole pairs. The MCuFe MOF heterojunction accelerates the cycling of ≡Fe^II^/≡Fe^III^ and Cu^II^/≡Cu^I^ in the photo-Fenton reaction system, ensuring the rapid regeneration of ≡Fe^II^ and the efficient production of ·OH. This strategy avoids leaching of metal ions by common catalysts under tightly controlled pH (2.8–3.5) by broadening the optimal pH range [73].

Feng et al. doped Cu^2+^ into Fe-BOD using the solvothermal method. The doping of Cu does not affect the original surface morphology but collapses the nanopores, changing the pore structure from micropores to mesopores. In this Fenton-like system, the Fe sites of FeCu (BDC-Br-1) tightly combine with H_2_O_2_ for electron transfer, generating ROS species (HO• and HO_2_•) in the Fe(III)/Fe(II) redox cycle reaction (Figure 8d). Within this reaction system, the four different free radicals contribute to the degradation of phenol in the following order: electron < ^1^O_2_ <  O_2_^–^ <  HO•. These findings suggest that HO• is the most prominent and important active species for this reaction. Analysis of the quasi-reduction potential shows that the reduction of Fe(III) by Cu(I) is thermodynamically favorable. Equations (1)–(3) demonstrate that the presence of Cu promotes the Fe(III)/Fe(II) redox cycle [74].
(1)$$\equiv {\mathrm{Fe}}^{Ⅲ}+e^{-}\to \equiv {\mathrm{Fe}}^{Ⅱ}$$
(2)$$\equiv {\mathrm{Cu}}^{Ⅱ}+e^{-}\to \equiv {\mathrm{Cu}}^{Ⅰ}$$
(3)$$\equiv {\mathrm{Cu}}^{Ⅰ}+\equiv {\mathrm{Fe}}^{Ⅲ}\to \equiv {\mathrm{Cu}}^{Ⅱ}+\equiv {\mathrm{Fe}}^{Ⅱ}$$

### 3.2. Bimetallic MOF for PMS Activation

The superior performance of PMS-AOPs in wastewater treatment has garnered considerable attention. This promising wastewater treatment method involves PMS activation by suitable MOFs to produce various reactive oxidizing substances that can attack a range of refractory organic pollutants [75].

In a study by Wang et al., as depicted in Figure 9a, 1,4-benzenedicarboxylic acid was selected as a ligand, and triethylamine (TEA) was used as a shape control agent to synthesize a two-dimensional FeCo-BDC nanosheet. Catalytic experiments demonstrated that the catalytic activity of bimetallic MOFs was superior to that of Fe-BDC and Co-BDC nanosheets. Doping Co-BDC with iron led to the occupation of guest ions within the internal space of MOFs, resulting in a slight reduction in specific surface area. Additionally, electron transfer between the bimetals effectively accelerated the redox process of Co/Fe, thereby promoting the formation of ROS. Finally, the main reactive oxygen species O_2_^•−^ and ^1^O_2_ decomposed dye molecules [76].

In another study, Gu et al. utilized a step-by-step synthesis method to prepare Fe and Co core–shell structure bimetallic MOFs (M/Z_2_) with MIL (Fe) as the core and a layer of ZIF-67 uniformly grown on the surface as the shell. This structural bimetallic MOF could effectively activate PMS in a short time to degrade high concentrations of 2-chlorophenol and several antibiotics. In core–shell bimetallic MOFs, MIL(Fe) acted as the core structure, providing high-performance Fe US, which enhanced the capture of pollutants by M/Z_2_. The primary mechanism of action involved the acceleration of the redox cycle of Co(II)/Co(III) by improving the internal electron transfer ability, further enhancing the excellent PMS activation ability of core–shell bimetallic MOF (Figure 9b). Both SO_4_^•−^ and •OH generated during the decomposition of PMS exhibit promotion effects on the degradation of 2-chlorophenol (2-cp). Benzoquinone intermediates such as 2-chloro-1,4-benzoquinone can activate PMS to ^1^O_2_, which plays a significant role in the degradation of 2-cp [77].

Several researchers have investigated the use of Co-doped Cu-MOFs for the degradation of nimesulide (NIM) in wastewater treatment and found that the degradation efficiency of CuCo-MOF was 7.3 and 2.4 times higher than that of Cu-MOF and Co-MOF, respectively (Figure 10a) [78]. The synthesis process of CuCo-MOF is shown schematically in Figure 8c. In situ characterization revealed that the pore structure of CuCo-MOF remained largely unchanged and had a similar specific surface area to Co-MOF. However, Co^2+^ doping greatly promoted electron transfer within MOFs and increased the number of active sites. X-ray photoelectron spectroscopy (XPS) demonstrated that the Cu 2p^+++^_3/2_ peak of CuCo-MOF shifted to a lower binding energy than that of Cu-MOF (Figure 10b). This may be due to the addition of Co promoting electron migration from the ligand to the Cu site via cation–π interactions, which increases the electron density of the metal site. Electrochemical impedance spectroscopy (EIS) (Figure 10c) also confirmed that the addition of Co to Cu-MOF crystals promoted interfacial electron transfer, demonstrating the synergistic effect of Co-Cu in promoting PMS activation. Finally, Figure 10d illustrates the adsorption energy of PMS on metal sites with different structures, with Co sites in CuCo-MOF displaying the strongest adsorption energy for PMS.

Zhang et al. [79] synthesized MIL-125(Ti) -NH_2_-Sal-Fe, a highly efficient photocatalytic material with strong photocatalytic activity. The photocatalytic mechanism is illustrated in Figure 11. Under illumination, the ligand-to-metal charge transfer process leads to the generation of photogenerated electrons and holes, which form Ti^3+^ and Fe^2+^. As an electron acceptor, PMS captures photoinduced electrons and produces SO_4_^•−^. In addition, dissolved oxygen in solution can also receive light-induced electrons to generate •O_2_^−^. Most of the SO_4_^•−^ generated by Fe^2+^ through PMS activation reacts with H_2_O to generate •OH. Therefore, in this photocatalytic system, three types of reactive oxygen species—SO_4_^•−^, •OH, and •O_2_^−^—act together on target compounds.

### 3.3. Bimetallic MOF for PDS Activation

PDS exhibits structural dissimilarities from PMS, resulting in different activation pathways and non-radical reactions. The symmetric molecular structure of PDS (^−^O_3_SO-OSO_3_^−^) renders it less nucleophilic for the attack of electron-rich organic pollutants compared to PMS (^−^O_3_SO-OH) [80].

Figure 12 illustrates that the morphology of the Co-doped material transitions from rod to sphere as the doping amount of Co^2+^ increases. When the Fe:Co ratio is 3:1, a spherical morphology emerges, as evidenced by the red circle highlighted in Figure. 12d. When assessing the impact of different ratios of Fe/Co on the catalytic activity of the material, a Fe/Co ratio of 1:1 was found to yield the strongest catalytic performance [81].

Sun et al. [82] synthesized MIL-88B (Fe/Co) using a hydrothermal method and mixed Co at various molar ratios for comparison. They discovered that Co doping affects the crystal surface morphology, with MIL-88B (Fe) transforming from its original spindle-like morphology to an irregular crystal morphology. Through methylene blue (MB) degradation tests, it was observed that Co^2+^ doping significantly enhanced the activation ability of PDS, particularly during the degradation period of 0–3 min. The incorporation of low-valence (Co^2+^) and high-valence (Fe^3+^) metals promoted electron transfer between the active site metals and amplified the activation ability of PDS, leading to an increase in the degradation efficiency (Figure 13a).

Zhang et al. [83] utilized PDS as an oxidant to evaluate the catalytic activity of FeCo/N-MOFs synthesized via a one-pot method. The FeCo/N-MOF/PS oxidation system achieved a high tetracycline (TC) removal rate (99.35%). SEM characterization of the materials revealed significant differences in the morphological structures of Co/N-MOF and FeCo/N-MOF. Figure 13b displays hexagonal spindle-shaped crystals with smooth surfaces, while Fe/N-MOF and Co/N-MOF exhibit octahedral crystals and flower-like structures, respectively (Figure 13c,d). This suggests that the competitive coordination between cobalt and iron sites and organic ligands modifies the structure of FeCo/N-MOF.

When oxidizing MOF with PMS, the composition and properties of the MOF may undergo changes. Some studies have shown that oxidizing PMS can convert some of the organic molecules on the surface of the MOF into functional groups such as carboxyl and hydroxyl groups, leading to the formation of some oxidation products on the MOF surface [84]. These oxidation products may alter the surface charge and hydrophilic properties of the MOF, affecting its adsorption and catalytic performance [85]. Furthermore, the oxidation process of PMS may lead to local damage to the MOF framework or alter the internal spatial structure of the MOF, which can affect its physical and chemical properties. When a MOF activates peroxide, the metal valence state inside the MOF can potentially change, depending on the type of metal ions present in the MOF and the reaction conditions. In general, peroxide can act as an oxidant and can transfer oxygen atoms to the metal ions in the MOF, leading to a change in their oxidation state.

## 4. Influence of Reaction Parameters on the Degradation Process

### 4.1. Effect of Bimetallic Stoichiometric Ratio

The stoichiometric ratio can significantly influence the adsorption and catalysis of materials by affecting their specific surface area and reactive sites. Excessive doping of metal ions can negatively impact the formation of porous structures. Several materials, including UTSA-16 [86], Ni-MOF-5 [87], MIL-101 (Cr, Mg) [88], MOF-5 [89] and HKUST-1 [90], have been reported to exhibit reduced particle surface area due to excessive metal doping. Moreover, the ratio of solute to solvent during the preparation process also significantly affects particle morphology. Yang Jimin et al. used a one-step synthesis method to dope Ni into MOF-5 and found that the volume ratio of C_2_H_5_OH: DMF could adjust the crystal size and shape by affecting the deprotonation rate of H_2_BDC and the growth rate of the n (100) and n (111) crystal planes. When the volume ratio of C_2_H_5_OH:DMF increased, the deprotonation rate of H_2_BDC slowed down, leading to a delay in the nucleation rate, which resulted in larger particles. According to the Bravais–Friedel–Donnay–Harker theory [91], the crystal structure is determined by the slower-growing crystal planes. When the volume ratio of C_2_H_5_OH:DMF is between 0 and 3:7, n (100) reacts slowly, and the crystal morphology is cubic. As the volume ratio of C_2_H_5_OH:DMF increases, the growth rate of the n (111) crystal plane approaches that of n (100), and the crystal shape becomes truncated octahedral [87].

### 4.2. Effect of pH Value of Reaction Solution

The pH of the reaction solution exerts a significant influence on multiple reaction activities such as oxidant decomposition and free radical generation, thereby affecting the degradation of pollutants [92]. For instance, PMS has pKa_1_ = 0.4 and pKa_2_ = 9.3. When the pH is below 9.3, the major form of PMS is HSO_5_^−^, which can be effectively activated to generate various ROS [93]. However, when the pH is too low, the excess H^+^ ions in the solution form stable hydrogen bonds with the O-O bonds in PMS, making it challenging to degrade PMS and reducing the production of active free radicals. Similarly, when the pH is above 9.3, the interaction between HSO_5_^−^ and hydroxide diminishes or transforms into SO_5_^2−^, which has lower activity, thereby significantly reducing the degradation effect [94]. Thus, extremely acidic and alkaline solutions are not conducive to pollutant degradation [95,96,97]. However, in a recent study by Debashis Roy [98], MIL-53 (Fe/Co)/CeO generated singlet oxygen and superoxide radicals in highly alkaline solutions that could attack unsaturated e-rich compounds (such as phenols, amines, and sulfides) and were not affected by anions in water, leading to a considerable enhancement of the overall degradation efficiency.

Fe-Mo@N-BC investigated by Yao et al. exhibited the highest catalytic activity for PMS under weak acidic conditions. As the pH increased from 2.74 to 10.17, the removal efficiency of pollutants gradually decreased from 100% to 61.1% [99]. The zeta potential measurement revealed that the surface of Fe-Mo@N-BC was negatively charged when the pH in the system was >3.05. This effect of electrostatic repulsion impeded the movement of PMS toward the material surface, thereby reducing the removal rate of Orange II.

### 4.3. Influence of Inorganic Anions

The catalytic performance of bimetallic MOF/PS systems in actual water bodies can be influenced by common water components such as Cl^−^, SO_4_^2−^, NO_3_^−^, HCO_3_^−^, CO_3_^2−^, and other anions [100]. Changes in the solution pH, ROS trapping, and neutralization of electrostatic forces between reactants can affect pollutant degradation and reduce catalyst performance [101,102]. Each anion has a unique chemical reactivity in the reaction system, leading to different effects on pollutant removal [103,104]. Xiao et al. observed that anions could interact with sulfate radicals and hydroxyl radicals to form species with low oxygen activity. Additionally, they increased the pH of the reaction solution, which inhibited the degradation of metronidazole (MNZ) [105] The degree of inhibition of these anions on the degradation of MNZ followed this order: HCO_3_^−^ > Cl^−^ > HPO_4_^2−^. Yao et al. reported that HCO_3_^−^ increased the pH of the reaction system and inhibited the degradation of Orange II [99]. Cl^−^ in natural water bodies can remove sulfate and hydroxyl radicals to form chlorine free radicals with low activity. Moreover, HCO_3_^−^ and CO_3_^2−^ generate HCO_3_^−^ and CO_3_^2−^ species with low oxidative properties, while H_2_PO_2_^−^ can bind to the active sites on Fe-Mo@N-BC and hinder the activation of PMS.

## 5. Regeneration and Stability of Bimetallic MOFs

During the activation of PMS, heterogeneous metal-based catalysts may undergo deactivation, leading to a decrease in catalytic activity over time. The formation of various intermediate products during the reaction can reduce catalyst activity and slow the rate of degradation. Moreover, leaching of metal ions is a common issue with metal-based catalysts, which can lead to secondary pollution of water bodies and a reduction in catalyst activity [106,107]. Material stability is crucial for the recovery and reuse of heterogeneous catalysts in the solution. Therefore, improving material stability is significant in heterogeneous catalysts.

Research has indicated that doping metals into MOFs can effectively enhance material stability [108]. For instance, ZIF-67 is known to be unstable in water, but after Yao et al. doped Zn into ZIF-67, the resulting ZnCo co-doped ZIF particles (with Co content of 25%, 50%, and 75%, respectively) maintained stable crystallinity and structure after 24 h in water. The presence of Zn(II) in the framework significantly improved the chemical stability of co-doped ZIF materials [43]. Additionally, Xiao et al. prepared bimetallic N-rich biochar Fe-Ce@N-BC and studied the leaching of iron in the Fe-Ce@N-BC/PMS system using ICP-MS. They found that Fe-Ce@N-BC doped with Ce had lower iron ion leaching compared to Fe@N-BC [99,105].

Some concerns about the environmental impact of MOFs include their potential toxicity to biological organisms, their potential to release metal ions or organic ligands into the environment, and their potential to adsorb or release pollutants. Ji et al. [109] found that some MOFs can adsorb pollutants in water, such as heavy metals and organic pollutants, which could be a potentially useful application in water treatment. However, some MOFs can release metal ions into water, which may be harmful to aquatic organisms and human health. Timothy et al. [110] use MOFs for carbon capture and storage, which may help reduce greenhouse gas emissions. Some MOFs may release carbon dioxide into the environment if not stored or disposed of properly. One potential impact of MOFs on soil is that they could help remediate polluted soils by adsorbing or sequestering pollutants. For example, MOFs have been used to remove heavy metals and organic pollutants from polluted soils, which can help reduce the impact of these pollutants on the environment [111]. However, MOFs can inhibit the growth of soil microorganisms, thereby negatively affecting soil health and ecosystem function.

Overall, the environmental impact of MOFs remains an area of active research, and more work is needed to understand their potential risks and benefits. While some MOFs may have promising applications in environmental remediation and mitigation, caution should be exercised to ensure their safe and responsible use, minimizing their impact on soil, water, and air.

## 6. Conclusions

This review provided a summary of the preparation methods of bimetallic MOFs, the process of their efficient activation of oxidants for removing organic pollutants, and the effects of reaction parameters on the degradation process. Through a systematic comparison of the structural characteristics and application processes of various bimetallic MOFs, it became clear that the development trend of bimetallic MOFs catalysts is to achieve efficient design and preparation of catalysts with high activation performance, continuous stability, controllable cost, and easy synthesis to enable effective degradation of organic pollutants.

In conclusion, bimetallic MOFs have exhibited significant potential for the degradation of pollutants by activating peroxides. However, there is a need for further research to investigate the underlying mechanisms of peroxide activation. Additionally, limitations in the recycling and reusability of bimetallic MOFs restrict their practical implementation on a larger scale. In the future, our research on bimetallic MOFs needs to focus on the following aspects:The problem of metal ion leaching in bimetallic MOFs needs to be addressed to prevent a decrease in catalyst activity. Thus, it is essential to explore the stable material structure of bimetallic MOFs.The recycling of bimetallic MOFs remains a major challenge. While most bimetallic MOFs achieve a high pollutant removal rate, their small particle size makes it difficult to recycle them, and the structure of the material itself leads to easy loss and other issues. Therefore, future research must focus on achieving high activation performance and easy and efficient recycling of bimetallic MOFs.Presently, the range of available bimetallic MOFs is restricted, and there are numerous unexplored combinations of metal ions remaining to be utilized in their synthesis. The development of novel materials is imperative to expand this repertoire.The activation mechanism of peroxide by bimetallic MOFs is intricate, and the precise control of its pathway presents a notable challenge. Consequently, further investigations into the underlying reaction mechanisms are necessary for future progress in this area.

## Figures and Tables

**Figure 1 molecules-28-03622-f001:**
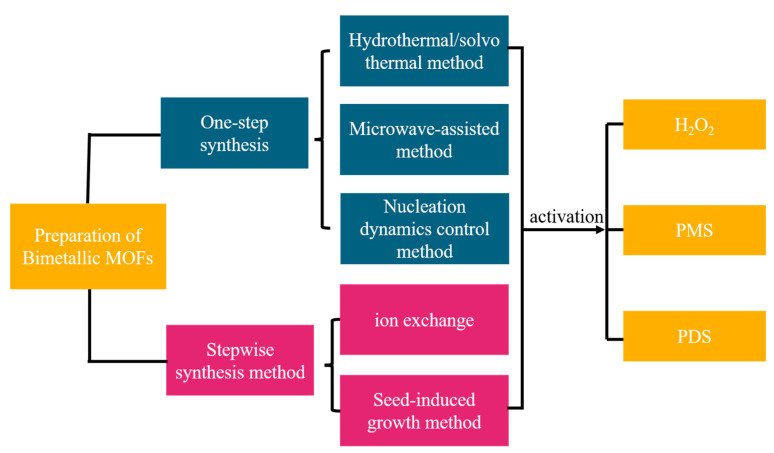
Research flowchart.

**Figure 2 molecules-28-03622-f002:**
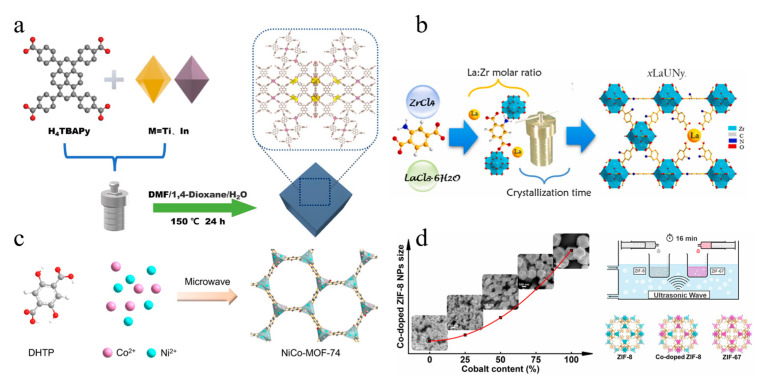
(**a**) Schematic representation of the synthesis of Ti-In-MOFs. Adapted with permission from Ref. [40], 2023, Elsevier; (**b**) La-Zr MOFs. Adapted with permission from Ref. [41], 2023, Elsevier; (**c**) NiCo-MOF-74. Adapted with permission from Ref. [42], 2023, Elsevier; (**d**) Co/ZIF-8. Adapted with permission from Ref. [43], 2023, Elsevier.

**Figure 3 molecules-28-03622-f003:**
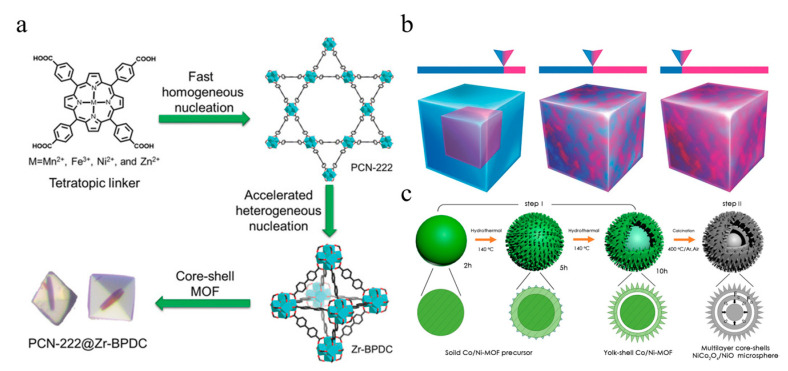
(**a**) Schematic diagram of the one-step synthesis of core-shell metal–organic framework hybrids: PCN-222@Zr-BPDC. Adapted with permission from Ref. [47], 2023, John Wiley and Sons; (**b**) Schematic illustration of Co_20_Zn_80_-ZIF, Co_50_Zn_50_-ZIF, and Co_80_Zn_20_-ZIF. Adapted with permission from Ref. [50], 2023, John Wiley and Sons; (**c**) Ni/Co-MOF. Adapted with permission from Ref. [51], 2023, Elsevier.

**Figure 4 molecules-28-03622-f004:**
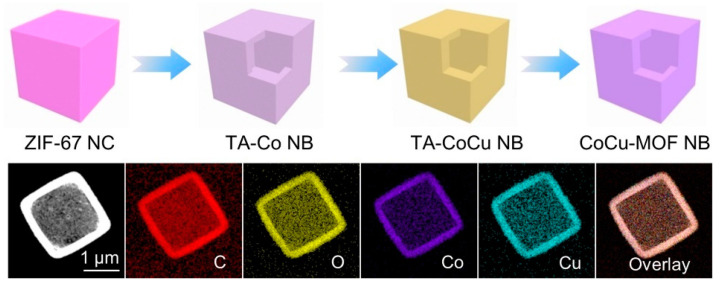
Schematic diagram of CuCo-MOF prepared by ion exchange method. Adapted with permission from Ref. [54], 2023, John Wiley and Sons.

**Figure 5 molecules-28-03622-f005:**
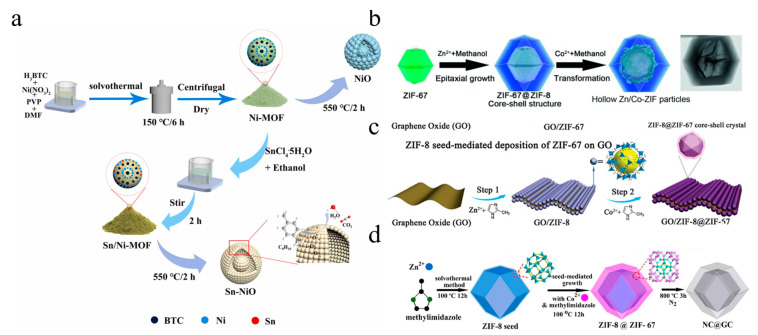
Schematic diagram of the stepwise synthesis of core–shell metal–organic framework hybrids. (**a**) Sn/Ni-MOF. Adapted with permission from Ref. [61], 2023, Elsevier; (**b**) Zn/Co bimetallic MOF. Adapted with permission from Ref. [62], 2023, John Wiley and Sons; (**c**) GO/ZIF-8@ZIF-67. Adapted with permission from Ref. [63]; (**d**) ZIF-8@ZIF-67. Adapted with permission from Ref. [64], 2023, American Chemical Society.

**Figure 6 molecules-28-03622-f006:**
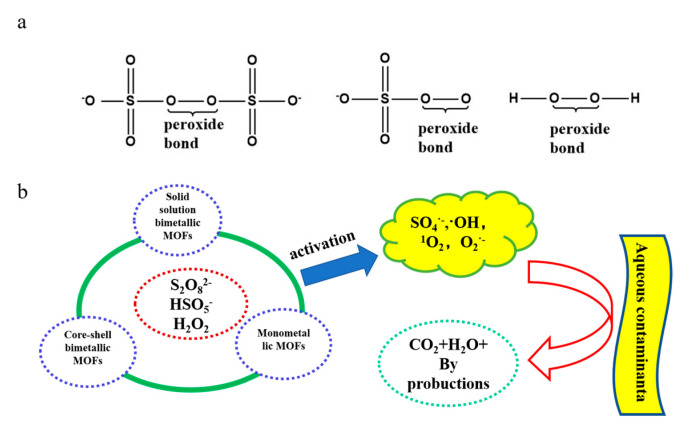
(**a**) Molecular structure of PDS, PMS, and H_2_O_2_; (**b**) Schematic diagram of the MOF activation PDS, PMS, and H_2_O_2_ process. Adapted with permission from Ref. [67], 2023, Elsevier.

**Figure 7 molecules-28-03622-f007:**
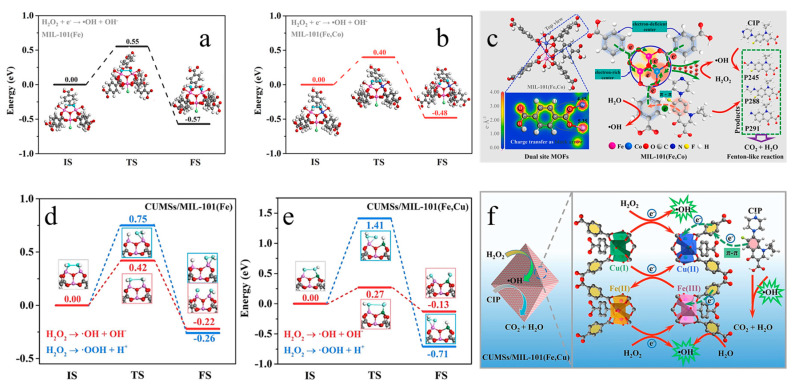
Calculated energy profiles for H_2_O_2_ dissociation on (**a**) MIL-101 (Fe) and (**b**) MIL-101 (Fe, Co) surfaces. (**c**) Schematic diagrams of the proposed mechanism involved in CIP degradation by MIL-101 (Fe, Cu)/H_2_O_2_. Adapted with permission from Ref. [71], 2023, Elsevier. Calculated energy profiles for H_2_O_2_ dissociation on (**d**) CUMSs/MIL-101(Fe) and (**e**) CUMSs/MIL-101 (Fe, Co) surfaces; (**f**) schematic diagrams of the proposed mechanism involved in CIP degradation by CUMSs/MIL-101 (Fe, Cu)/H_2_O_2_. Adapted with permission from Ref. [72], 2023, Elsevier.

**Figure 8 molecules-28-03622-f008:**
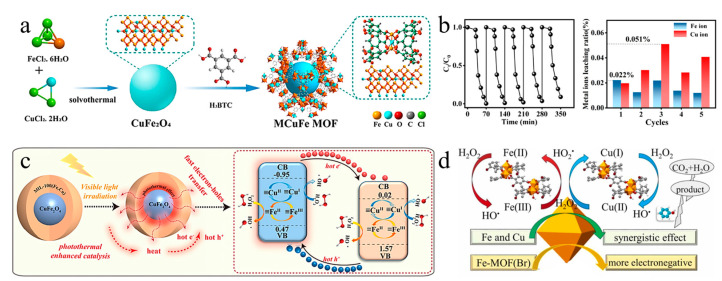
(**a**) Schematic diagram of MCuFe-MOF synthesis; (**b**) recycling and ion leaching of MCuFe-MOF; (**c**) Fenton reaction mechanism of MCuFe MOF glazing under visible light irradiation. Adapted with permission from Ref. [73], 2023, Elsevier. (**d**) Schematic diagram of the Fenton reaction mechanism of FeCu (BDC-Br-1). Adapted with permission from Ref. [74], 2023, Elsevier.

**Figure 9 molecules-28-03622-f009:**
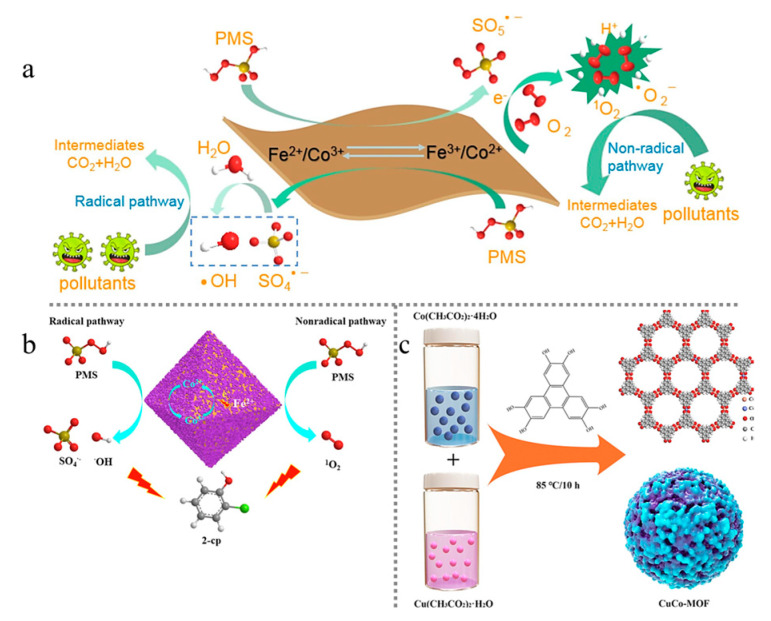
Schematic diagram of the reaction mechanism of (**a**) FeCo-BDC activated PMS (adapted with permission from Ref. [76], 2023, Elsevier) and (**b**) M/Z_2_ activated PMS (adapted with permission from Ref. [77], 2023, Elsevier). (**c**) Schematic diagram of CuCo-MOF synthesis process. Adapted with permission from Ref. [78], 2023, Elsevier.

**Figure 10 molecules-28-03622-f010:**
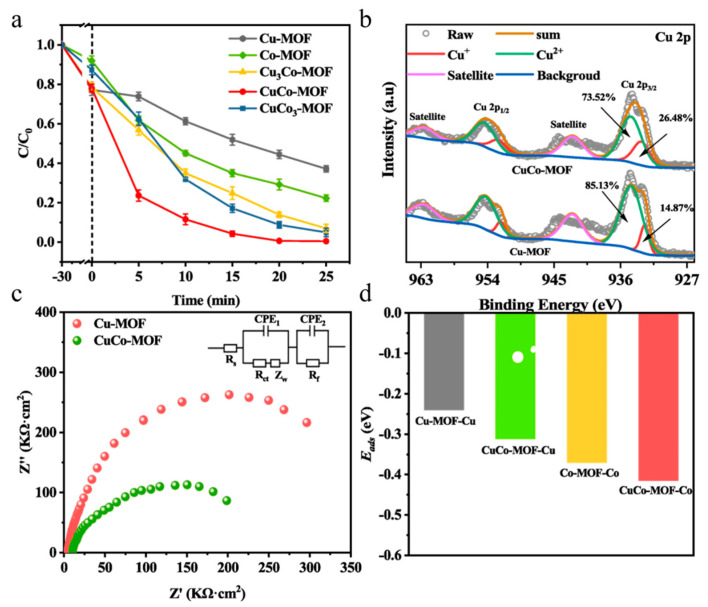
(**a**) Degradation efficiency curves of different materials for NIM; (**b**) Cu 2p XPS spectra of CuCo-MOF and Cu-MOF; (**c**) Nyquist plots of CuCo-MOF and Cu-MOF; and (**d**) adsorption energy of PMS on metal sites of different structures. Adapted with permission from Ref. [78], 2023, Elsevier.

**Figure 11 molecules-28-03622-f011:**
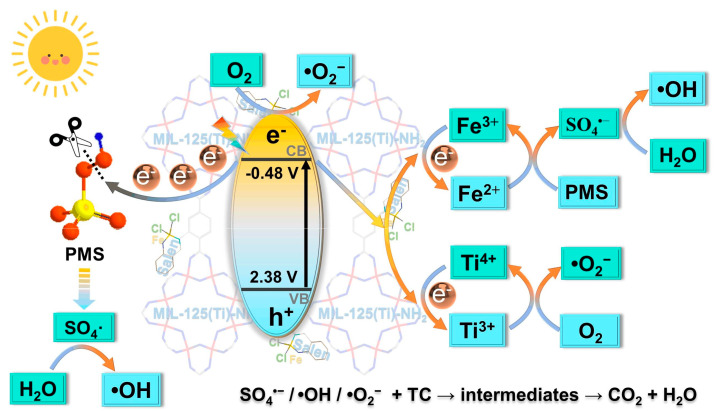
Mechanism for TC degradation by MSF/PMS/light system. Adapted with permission from Ref. [79], 2023, Elsevier.

**Figure 12 molecules-28-03622-f012:**
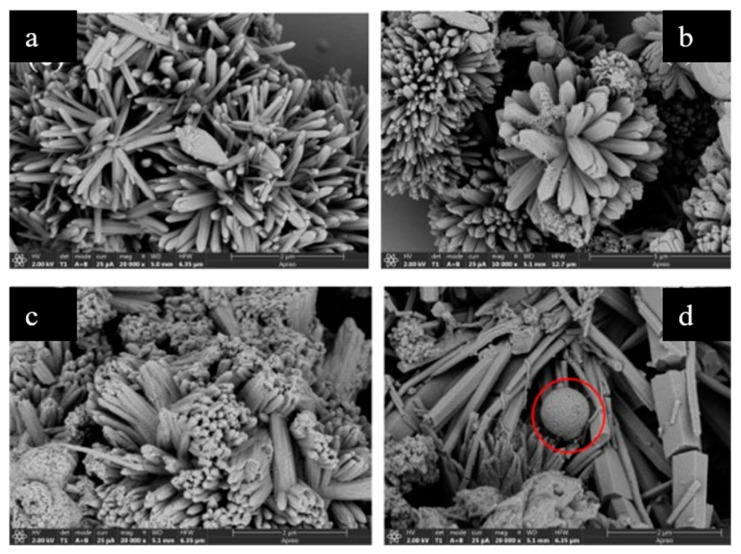
SEM images of (**a**) Fe_1_Co_3_-MOF-74, (**b**) Fe_1_Co_2_-MOF-74, (**c**) Fe_2_Co_1_-MOF-74, and (**d**) Fe_3_Co_1_-MOF-74. Adapted with permission from Ref. [81], 2023, Elsevier.

**Figure 13 molecules-28-03622-f013:**
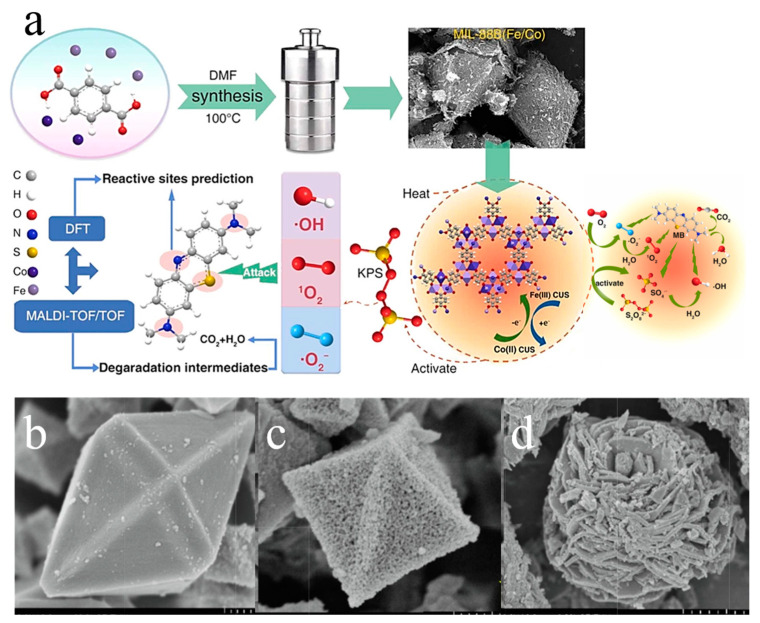
(**a**) Synthesis process of MIL-88B (Fe/Co) and heterogeneous activation mechanism of PDS. Adapted with permission from Ref. [82], 2023, Elsevier. (**b**) SEM images of FeCo/N-MOF, (**c**) Fe/N-MOF, and (**d**) Co/N-MOF. Adapted with permission from Ref. [83], 2023, Elsevier.

**Table 1 molecules-28-03622-t001:** Activation of MOF materials by different peroxides.

Type of Catalyst	Synthesis Method	Peroxides	Mechanism	Performance	Cycle Times	Reference
MIL-101 (Fe, Co)	Hydrothermal	H_2_O_2_	C-O-Fe/Co bonds and Fe-O-Co bonds facilitate electron transfer; π-π interaction between CIP and MIL-101 (Fe, Co)	30 min, 97.8% (CIP = 20 mg/L, catalyst = 0.2 g/L, H_2_O_2_ = 5 mM)	——	68
CUMS/MIL-101 (Fe, Cu)	Hydrothermal	H_2_O_2_	π–Cation interactions; favorable reaction between Cu(I) and Fe(III)	30 min, 100% (CIP = 20 mg/L, catalyst = 0.1g/L, H_2_O_2_ = 3 mM)	——	69
MCuFe MOF	Hydrothermal	H_2_O_2_	Holes and electrons can be heated into “hot electrons” and “hot holes”	40 min, 95% (MB =50 mg/L, catalyst = 0.05 g/L, H_2_O_2_ = 5 mM)	——	70
FeCu (BDC-Br)	Hydrothermal	H_2_O_2_	Fe-Cu electron transfer process promotes the decomposition of H_2_O_2_	60 min, nearly 100% (phenol =100 mg/L, catalyst = 0.1 g/L, H_2_O_2_ = 8 mM)	4	71
FeCo-BDC	Hydrothermal	PMS	Redox cycle between Co^3+^/Co^2+^ and Fe^3+^/Fe^2+^ promotes PMS activation	5 min, 99.1% (RhB = 20 mg/L, catalyst = 20 mg/L, PMS = 0.25 mM)	——	72
Fe-Co MOFs	Nucleation dynamics control	PMS	Synergy of cobalt and iron active sites promotes redox cycling of Co^2+^/Co^3+^	30 min, 90.3% (2-cp = 100 mg/L, catalyst = 0.1g/L, PMS = 0.3 g/L)	5	73
CuCo-MOF	Hydrothermal	PMS	Synergistic effect of Cu and Co facilitates electron transfer from electron-rich regions to metal sites	25 min, 100% (NIM = 20 mg/L, catalyst = 0.2 g/L, PMS = 3 mM)	——	74
MIL-88B (Fe/Co)	Hydrothermal	PDS	Co^2+^ can accelerate electron transfer and promote the cycle of Fe^3+^ and Fe^2+^	30 min, 99.85% (MB = 0.1 mM, catalysts = 0.5 g/L, PDS = 10 mM)	4	75
FeCo/N-MOF	Hydrothermal	PDS	Interaction of Fe(II) and Co(III) promotes the cycling of Co(II)/Co(III) and Fe(II)/Fe(III); doped N helps to generate ^1^O_2_	3 h, 98.60% (TC = 50 mg/L, catalyst = 0.2 g/L, PDS = 5 mmol/L)	——	76

## Data Availability

No new data were created or analyzed in this study. Data sharing is not applicable to this article.

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
