# Peer review of "Synthesis and Peroxide Activation Mechanism of Bimetallic MOF for Water Contaminant Degradation: A Review"

_molecules, 2023, doi:10.3390/molecules28083622_

Round 1
Reviewer 1 Report
Minor revision is needed as follow:
1. The title of the paper involves mechanisms, but the mechanism part of the paper is not emphasized. Therefore, the author should consider revising the title.
2. As for a review paper, the latest papers related to hydrogen peroxide activation is recommended for authors to read and cite
-(Catalysts 2023, 13, 10. https://doi.org/10.3390/catal13010010);
-(Separation and Purification Technology 312 (2023) 123452. https://doi.org/10.1016/j.seppur.2023.123452
3. The author should provide a table summarizing a comparison of hydrogen peroxide activation by different MOF materials.
4. The author should increase the discussion of the role of free radicals to highlight the mechanism part.
Reviewer 2 Report
1- Photocatalysis process in large scale applications should be highlighted in the introduction part. Please, see, (https://doi.org/10.1016/j.jwpe.2022.102847) ; (https://doi.org/10.1016/j.cej.2018.09.167).
2- There is no proper research methodology section portrayed in this study. The author should at least have a research flowchart that portrays the several stages in this study.
3- Add more descriptive figures showing the photocatalytic mechanism of the discussed materials/composites.
4- The authors should list and compare more recent studies highlighting the conducted preparation method, environmental conditions and removal efficiency in a summarized table.
5- What is the impact of using these materials/composites on the environment (Soil, Water and Air)? Revise in an additional section.
6- The authors have to consider the copy rights of all figures.
7- Figures from 1 to 7 are poor and should be regenerated (increase font size and resolution).
8- The authors should list and compare more recent studies about material regeneration and stability in a summarized table.
9- An additional section about the common sludge composition and characteristics should be added.
10- The authors should identify the limitations and future prospects of this review in an additional section.
11- The recommended future studies should be added in the conclusion part.
12- Bibliography part must be updated.
Round 2
Reviewer 2 Report
Accept.